# Phosphoproteomic Profiling Reveals mTOR Signaling in Sustaining Macrophage Phagocytosis of Cancer Cells

**DOI:** 10.3390/cancers16244238

**Published:** 2024-12-19

**Authors:** Bixin Wang, Xu Cao, Krystine Garcia-Mansfield, Jingkai Zhou, Antigoni Manousopoulou, Patrick Pirrotte, Yingyu Wang, Leo D. Wang, Mingye Feng

**Affiliations:** 1Department of Immuno-Oncology, Beckman Research Institute, City of Hope, Duarte, CA 91010, USA; 2Cancer and Cell Biology Division, Translational Genomics Institute, Phoenix, AZ 85004, USA; 3Integrated Mass Spectrometry Shared Resource, City of Hope Comprehensive Cancer Center, Duarte, CA 91010, USA; 4Center for Informatics, City of Hope, Duarte, CA 91010, USA; 5Department of Pediatrics, City of Hope National Medical Center, Duarte, CA 91010, USA

**Keywords:** macrophage, mTOR, phosphoproteomics, cancer immunotherapy

## Abstract

Investigating the phagocytic activity of tumor-associated macrophages (TAMs) provides valuable insights into the understanding of cancer immune evasion mechanisms and the development of innovative strategies for cancer therapy. However, the intracellular signaling networks that drive and sustain macrophages phagocytosis of cancer cells remain poorly understood. In this study, we elucidate the mechanisms underlying the sustentation of macrophage-mediated phagocytosis of cancer cells through a comprehensive phosphoproteomic analysis. Importantly, we uncover the critical role of mTOR signaling in regulating the phagocytic capacity of macrophages toward cancer cells via controlling phagosome maturation and mediating the transition between non-phagocytic and phagocytic states of macrophages in the presence of cancer cells.

## 1. Introduction

Tumor-associated macrophages (TAMs) are often the most abundant immune cell groups within the tumor microenvironment. They play a pivotal role in the tumor microenvironment (TME), contributing to tumor angiogenesis, extracellular matrix remodeling, cancer cell proliferation, tumor invasion and metastasis, and immunosuppression [1,2,3]. Comprising multiple phenotypically distinct subpopulations, TAMs can directly phagocytose malignant cells, irrespective of their polarization status [4,5]. Consequently, targeting the cancer–TAM interaction to induce phagocytosis has emerged as a promising therapeutic strategy for various cancers [6,7,8,9].

Macrophage-mediated phagocytosis is a multi-step process, encompassing target cell detection and recognition, cellular engulfment, and subsequent digestion of the target cells within phagolysosomes [6,10,11]. Several anti-phagocytic “don’t eat me” signals, such as CD47, CD24, and MHC I, have been identified, the blockade of which can unleash phagocytosis toward cancer cells [12,13,14,15]. Conversely, “eat me” signals such as calreticulin-binding sialoglycans on cancer cells, expressed due to oncogenic stress or paracrine decoration by macrophages, facilitate the recognition of target cancer cells [16,17,18]. Phagosome acidification in the macrophages, driven by V-ATPases that pump protons into the phagosomal lumen, is crucial for antimicrobial defense, enzymatic activation, and phagosome maturation [19,20]. While significant efforts have been devoted to identifying the signals that mediate cancer–macrophage interactions to initiate phagocytosis, the mechanisms that regulate its sustentation and resolution remain obscure. The mechanisms responsible for propagating pro-phagocytic signaling within macrophages have not been thoroughly explored, nor have the key signaling events for post-phagocytic responses been identified. A better understanding of the molecular mechanisms governing phagocytosis of cancer cells by macrophages is essential for uncovering novel therapeutic targets and strategies to enhance cancer treatment efficacy.

Protein phosphorylation and dephosphorylation are reversible post-translational modifications that are pivotal to the activation and inhibition of signal transduction, cell proliferation, and cell differentiation [21,22,23]. Phosphorylation is meticulously regulated by the opposing activities of protein kinases and phosphatases, which modify the phosphorylation of serine, threonine, and tyrosine residues [21,24]. Macrophage-mediated tumor cell phagocytosis is regulated by Src family kinases and phosphatases such as SHP1 and SHP2, which transduce the signals to activate or inhibit phagocytosis [25,26,27,28]. However, the dynamic regulation and functional impact of protein phosphorylation during tumor cell phagocytosis remain unclear. Identifying such mechanisms is essential for developing innovative macrophage-based cancer immunotherapies.

mTOR (the mechanistic target of rapamycin) is a serine/threonine kinase that belongs to the phosphatidylinositol 3-kinase-related kinase protein family [29]. The mTOR pathway is frequently dysregulated in cancer, making its inhibition a promising therapeutic strategy [30,31,32]. mTORC1 promotes protein synthesis by phosphorylating two key effectors: S6K1, which directly aids translation, and 4EBP, which inhibits translation by binding to eIF4E and preventing the assembly of the eIF4F complex [33,34]. mTORC2 primarily functions by phosphorylating members of the AGC (PKA/PKG/PKC) family of protein kinases or Akt [35,36,37]. mTORC1 and mTORC2 complexes are involved in various immune activities. Crosstalk between mTORC1 and mTORC2 is essential for NK maturation and effector functions [38]. Additionally, mTORC1, but not mTORC2, is crucial for the differentiation of follicular regulatory T cells [39]. mTORC1 signaling also regulates proinflammatory macrophage function, differentiation, and metabolism [40,41,42]. Constitutive mTORC1 activity impairs M2 polarization [42], while mTORC2 signaling regulates M2 macrophage differentiation [43]. Thus far, the role of the mTOR pathway in the macrophage-mediated phagocytosis of cancer cells remains inadequately defined.

Mass-spectrometry-based phosphoproteomics has emerged as a powerful tool for investigating comprehensive cellular signaling networks [44,45]. Here, through a longitudinal analysis of the macrophage phosphoproteome during tumor phagocytosis, combining tandem-mass-tag (TMT) labeling and two-dimensional liquid chromatography–tandem mass spectrometry (LC-MS/MS) with KEGG pathway analysis, we have elucidated the critical phosphorylation signaling networks within macrophages that regulate tumor cell phagocytosis. Our findings reveal a temporally dynamic phosphoproteome that is distinct between the initiation and sustentation phases of macrophage phagocytosis. The KEGG analysis highlights the activation of the mTOR pathway following the initiation of cancer cell phagocytosis. Functional investigation demonstrates that the inhibition of mTOR signaling significantly impairs the phagocytic capacity of the macrophages without impacting their affinity with phagocytosis. Compared to previous research, which has broadly examined the changes in phosphorylation associated with macrophage activation or polarization mechanisms [46,47,48], our study specifically focuses on macrophages’ phagocytic pathways—an essential aspect of innate-immunity-mediated cancer immunosurveillance that remains underexplored at the phosphoproteome level. Importantly, we identify previously uncharacterized phosphorylation sites involved in macrophage phagocytosis and provide a comprehensive view of its dynamic regulation. Moreover, we uncover the novel regulatory role of mTOR signaling in phagosome acidification in the macrophages, a process critical to the sustentation of phagocytosis and the transition of macrophages from non-phagocytic to phagocytic states, revealing a distinct mechanism in the regulation of phagocytosis of cancer cells by macrophages. Manipulating mTOR signaling offers a promising strategy for enhancing cancer cell phagocytosis, paving the way for innovative macrophage-based therapeutic approaches. Identifying and understanding such mechanisms not only shed light on the fundamental principles of cancer cell phagocytosis but are also pivotal in inspiring the development of innovative macrophage-based cancer immunotherapy approaches.

## 2. Materials and Methods

### 2.1. Generation and Culturing of the Macrophages

#### 2.1.1. Bone-Marrow-Derived Macrophages (BMDMs)

BMDMs were generated from six- to ten-week-old BALB/c mice, as described previously [49,50]. Bone marrow cells were flushed from the femurs using a 25 G needle and filtered through a 70 μm strainer. The cells were then pelleted through centrifugation, and red blood cells were lysed with ACK lysis buffer at room temperature for 3 min. After washing them with IMDM supplemented with 10% FBS, the cells were cultured in IMDM containing 10% FBS and 10 ng/mL murine M-CSF. BMDMs from day 6 to day 8 were used for the functional assays in this study.

#### 2.1.2. Human Macrophages

Human macrophages were generated by enriching monocytes from the peripheral blood using magnetic-activated cell sorting with whole blood CD14 microbeads (Miltenyi), as described previously [49,50]. CD14^+^ monocytes were cultured in IMDM supplemented with 10% human serum. Human macrophages from day 6 to day 8 were used for this study.

### 2.2. Cell Culture

Raji (CCL-86), SW620 (CCL-227), and MDA-MB-231 (HTB-26) tumor cells (all from the ATCC) were cultured in their respective media with 10% FBS and 1% penicillin/streptomycin. The Raji cells were grown in RPMI medium, while the SW620 and MDA-MB-231 tumor cells were cultured in DMEM. The cells were maintained at 37 °C in a humidified environment containing 95% air and 5% CO_2_. Regular screening was performed to ensure the absence of mycoplasma contamination.

### 2.3. Co-Culture of the Macrophages with Human Cancer

The BMDMs were co-cultured with human cancer Raji cells for 0, 1, 15, 60, and 120 min in the presence of a CD47-blocking antibody (B6H12), with biological replicates. After the designated incubation periods, the Raji cells were washed away, and the macrophages were collected for phosphoproteome analysis.

### 2.4. Sample Preparation for Phosphoproteomic Profiling

Samples were dissolved in 100 µL lysis buffer (0.5M triethylammonium bicarbonate, 0.05% sodium deoxycholate). The samples were subjected to tip sonication (Q700, QSonica; amplitude = 10; 2 s on/2 s off pulses; 20 s total processing time per sample; on ice) and then centrifuged at 15K rpm at 4 °C for 10 min. The supernatant was transferred into a fresh tube (on ice), and the protein concentration of each sample was measured using the Pierce Bradford Plus Protein Assay Reagent (Thermo Fisher Scientific, Waltham, MA, USA; #23238) per the manufacturer’s instructions. An equal amount of protein (100 µg) per sample was transferred into a fresh tube and adjusted to the highest volume with lysis buffer. Then, 4 µL of Reducing Reagent (Sigma-Aldrich, St. Louis, MI, USA; #4381664) was added to each sample. The samples were incubated at 60°C for 1 h. Then, 2 µL of Alkylating Reagent (Sigma-Aldrich, St. Louis, USA; #4381664) was added to each sample. The samples were incubated at room temperature for 15 min. Then, 4 µg trypsin/LysC (Promega, Madison, WI, USA; #V50703) was added to each sample. The samples were incubated overnight at room temperature in the dark.

TMT-11plex reagents (Thermo Fisher Scientific, Waltham, MA, USA; #A37725) were equilibrated at room temperature. Then, 41 µL of anhydrous acetonitrile (Sigma, 900644) was added to each label, and the contents were transferred into each sample. The samples were incubated at room temperature for 1 h. A total of 8 µL of 5% hydroxylamine was added to each sample. The samples were incubated at room temperature for 15 min and were then combined and dried using a SpeedVac (Eppendorf 5301 vacufuge concentrator). We performed phosphopeptide enrichment on the pooled TMT-labeled samples by sequentially using a High Select TiO_2_ phosphopeptide enrichment kit (Thermo Fisher Scientific, Waltham, MA, USA; #A32993), followed by a High Select Fe-NTA phosphopeptide enrichment kit (Thermo Fisher Scientific, Waltham, MA, USA; #A32992) per the manufacturer’s instructions. The TiO_2_- and Fe-NTA-enriched phosphopeptide fractions were acquired separately.

### 2.5. Mass Spectrometry

Mass spectrometry (MS) data acquisition was carried out using a U3000 RSLCnano liquid chromatography system (Thermo Fisher Scientific, Waltham, MA, USA) connected to an Orbitrap Fusion Lumos Tribrid mass spectrometer (Thermo Scientific). Tryptic phosphopeptides were loaded onto a 50 cm C18 column and separated using a 300 min RP chromatography gradient. The eluted peptides were directly nano-sprayed, and the spectra were collected using a data-dependent acquisition method. Briefly, MS1 scans were acquired across a scan range of 400–1600 *m*/*z* in the Orbitrap at a 240K resolution, followed by MS2 scans on the most abundant precursors in the Orbitrap at a 50K resolution. Only precursors with charge states between 2 and 7 were selected for HCD MS2 scans. To prevent resampling of the same precursors, a dynamic exclusion setting of 60s was applied.

### 2.6. Protein Identification and Data Analysis

The raw spectra were searched against a mouse database (Swissprot/UniprotKB, Mus musculus, downloaded in August 2020) using Byonic (v2.16) run through Proteome Discoverer v2.4. The precursor mass tolerance was set to 10 ppm and the fragment mass tolerance to 20 ppm, with HCD fragmentation. Phosphorylation (STY), oxidation (M), and deamidation (NQ) were set as variable modifications, with methylthio (C) and TMT6plex (N-term, K) as fixed modifications. A maximum of 4 modifications were allowed per peptide. PSMs with <1% FDR were retained for peptide and protein quantification.

Phosphosite abundance was normalized first through internal reference scaling, followed by a sample-loading method, as described by Wilmarth. To reduce the phosphopeptides to the phosphosite level, unique phosphosites were identified through global protein localization, and the abundance for all peptides containing a given phosphosite was averaged [51].

To identify differentially abundant (DA) phosphosites over time, we utilized the lmFit and eBayes functions from the limma package in R v3.5.2. *p*-values were adjusted with Benjamini–Hochberg correction for multiple testing (q-value) and filtered for a q-value < 0.05 for significance [52].

The clustering of the phosphosite abundance patterns over time was calculated via the degPatterns function in DEGReport and filtered for clusters with >50 phosphosites. The phosphosites in each cluster were then reduced to their gene identity, mapped to their KEGG id counterparts, and analyzed against a mouse database via enrichKEGG in clusterProfiler [53]. All plots were generated in R v4.2.2 using the ggplot2 or pheatmap packages.

The kinase activities were calculated for each sample using IKAP within MATLAB (v R2022a) [54]. In summary, all phosphosites, along with their normalized abundance in each sample, were input into IKAP, along with a list of known substrate–kinase pairs from PhosphoSite Plus for mice (downloaded August 2023). IKAP then calculated the relative kinase activity for each kinase that mapped to a phosphosite substrate in our dataset. These activities could then be compared across time points using the same time series method in limma as described above.

### 2.7. A Luminescence-Based Long-Term Macrophage Phagocytosis Assay

The Raji, SW620, and MDA-MB-231 cancer cell lines were transduced with a luciferase–eGFP fusion protein and co-cultured with the control or rapamycin-pretreated macrophages for a certain time frame in IMDM (10% FBS) in the presence or absence of the CD47-blocking antibody. After incubation, luciferin was added into each well using a multichannel pipettor, followed by the detection of luminescence signals using the Cytation 3. A tumor-only group without macrophages was used as a normalization control for calculation, in which the phagocytosis rate was 0%. The phagocytosis rate was quantified as the ratio of the signals in the treatment group to the signals of the cancer cells only.

### 2.8. The Phagocytosis Assay Using Flow Cytometry or Microscopy

#### 2.8.1. Flow Cytometry

The Raji cancer cells were labeled with CellTrace CFSE by incubating them in PBS for 30 min at 37 °C and then washed with RPMI and PBS. CFSE-labeled Raji cells were co-cultured with macrophages pretreated with or without rapamycin, in the presence of the CD47-blocking antibody, for a specified duration at 37 °C. Phagocytosis was terminated by washing the cells with PBS. The macrophages were stained with an anti-mouse F4/80 antibody for flow cytometry. Phagocytosis was quantified as the percentage of F4/80^+^ macrophages that phagocytosed the CFSE^+^ cancer cells.

#### 2.8.2. Microscopy

To obtain microscopic images of the phagocytosis assays, the same cell co-culture method was used as that in flow cytometry. After phagocytosis was completed, fluorescence images were taken using a Zeiss Observer Z1 Live Cell widefield microscope with a 20×/0.5 objective and a 1.6× optovar.

### 2.9. LysoTracker Staining for Flow Cytometry

The CFSE-labeled Raji cells were incubated with the control BMDMS or those pretreated with 100 nM rapamycin in the presence of the CD47-blocking antibody at the indicated concentrations for 2 h, 6 h, 12 h, or 24 h. After the termination of phagocytosis, the cells were then incubated with prewarmed IMDM with a 50 nM LysoTracker (Invitrogen, Thermo Fisher Scientific, Waltham, MA, USA) probe for 30 min under growth conditions. After incubation, the LysoTracker probe was removed, and the cells were washed with PBS. The macrophages were stained with the anti-mouse F4/80 antibody and then analyzed using flow cytometry.

### 2.10. The Rechallenge Experiment

The macrophages were seeded into 24-well Petri dishes and pretreated with 100 nM rapamycin for 24 h or left untreated. The medium was then removed, and CFSE-labeled Raji cells were added at a 1:3 ratio with either 0.02 or 0.2 µg/mL of the CD47-blocking antibody for 6 or 12 h. Afterward, the CFSE-labeled Raji cells were removed, and Far Red-labeled Raji cells at the same ratio, with the same concentrations of the CD47-blocking antibody, were added for an additional 18 or 12 h. At 24 h, phagocytosis was halted, and the macrophages were stained with the anti-mouse F4/80 antibody and analyzed using flow cytometry.

### 2.11. Chemicals and Cell Culture Reagents

Commercial reagents were obtained from the following suppliers: fetal bovine serum (FBS; Omega Scientific, Tarzana, CA, USA; #FB-02); cell culture media and additives (Gibco, Thermo Fisher Scientific, Waltham, MA, USA); penicillin/streptomycin (GenClone, Genesee Scientific, San Diego, USA; #25-512); murine macrophage colony-stimulating factor (M-CSF; Thermo Fisher Scientific, Waltham, MA, USA; #315-02-100UG); InVivoMAb anti-human CD47 B6H12 (Bio X Cell, Lebanon, PA, USA; #BE0019-1); rapamycin (LC Laboratories, Woburn, USA; #NC9362949); temsirolimus (Cayman Chemical, Ann Arbor, USA; #11590); D-Luciferin (Syd Labs, Beverly, CA, USA, #MB000102-R70170); TrypLE Express (Gibco, Thermo Fisher Scientific, Waltham, MA, USA; #12604-021); Sytox Blue buffer (Invitrogen, Thermo Fisher Scientific, Waltham, MA, USA; #S34857); 10× PBS(Corning, Corning, NY, USA; #46-103-CM); Annexin V binding buffer (BD Biosciences, San Jose, CA, USA; #556454); human serum (Omega Scientific, Tarzana, USA; #HS-20); LysoTracker (Invitrogen, Thermo Fisher Scientific, Waltham, MA, USA; #L7528); CellTrace CFSE (Invitrogen, Thermo Fisher Scientific, Waltham, MA, USA; #C34554); and CellTrace Far Red (Invitrogen, Thermo Fisher Scientific, Waltham, MA, USA; #C34564).

### 2.12. Flow Cytometry and Antibodies

For analysis of their surface markers, the cells were stained with a surface antibody mixture in FACS buffer (1× PBS containing 1% BSA + 1%NaN_3_) at 4 °C for 15 min. The cells were then washed with FACS buffer and suspended with Sytox Blue buffer (1:20,000). Cytometry data were acquired on Fortessa (BD Biosciences) or Aurora equipment and analyzed using Flowjo software (v10.9.0).

The antibodies used to stain the murine cells included anti-F4/80 (BM8), anti-CD80 (16-10A1), anti-CD86 (GL-1), anti-iNOS (W16030c), anti-MHC II (M5/114.15.12), anti-CD206 (C068C2), anti-FcүRI (X54-5/7.1), anti-FcүRIIB (AT130-2), anti-FcүRIII (93), anti-FcүRIV (9E9), and anti-Sirpα (P84).

### 2.13. The Cell Viability Assay

The macrophages treated with 100 nM rapamycin or the control macrophages were collected and washed with PBS. The cells were then stained with anti-F4/80 and washed again with PBS, followed by Annexin V binding buffer. Next, the cells were resuspended in a cocktail containing Annexin V (BD) and Annexin V binding buffer (BD) and incubated in the dark at room temperature for 20 min. The cells were suspended with Sytox Blue and analyzed using flow cytometry. Viable cells were identified as Annexin V-/Sytox Blue-populations.

### 2.14. CRISPR-Cas9-Mediated Gene Editing

The CRISPR-Cas9 system was utilized to knock down the gene expression in the primary macrophages isolated from CRISPR-Cas9 knock-in mice [Gt(ROSA)26Sorem1.1(CAG-cas9*,-EGFP)Rsky] [55], which constitutively expressed the Cas9 protein. In particular, control sgRNA (AGUCCGGUCGAAAUCUGUAU) or sgRNA targeting mouse MTOR (UGAUACGAACUAGCUCGUUG) [56,57] was cloned into the pLKO5.sgRNA.EFS.tRFP vector to deliver sgRNA independently of Cas9. This vector, provided by B. Ebert (Addgene, Watertown, NY, USA; plasmid #57823) [58], was used for sgRNA delivery. The lentiviral particles were concentrated following the established protocols [59].

### 2.15. Single-Cell RNA Sequencing Analysis

Single-cell RNA sequencing (scRNA-seq) data from the control groups, sourced from the GEO database (GSE180296 and GSE139492 [60,61]), were analyzed. The data were preprocessed using the Seurat package in R. A Principal Component Analysis (PCA) and Uniform Manifold Approximation and Projection (UMAP) were used for dimensionality reduction and visualization. Clustering was performed using the Louvain algorithm, with the clusters annotated based on known immune cell markers or tumor cell markers. Dot plots were used to visualize the expression of the immune cell markers or tumor cell markers. Additionally, the addModuleScore function in Seurat was used to calculate the mTOR pathway enrichment score.

### 2.16. Accession Numbers

The mass spectrometry proteomics data were deposited into the ProteomeXchange Consortium via the PRIDE partner repository, with the dataset identifier PXD053542 [62].

### 2.17. Statistical Analysis

The results presented in this study were obtained from at least two replicate experiments. The data are expressed as means ± SDs. Significant differences between groups were assessed using an unpaired t-test or an ANOVA (one-way or two-way). The statistical analysis was performed with Prism 10 GraphPad software, with the significance set at *p* < 0.05 (* *p* < 0.05, ** *p* < 0.01, *** *p* < 0.001, and **** *p* < 0.0001).

## 3. Results

### 3.1. Multiplexed Quantitative Analysis of the Phosphoproteome Is Applied to Analyzing Phagocytosis of Cancer Cells by Macrophages

To identify the phosphorylation events in the macrophages during the phagocytosis of the cancer cells, we employed SMOAC enrichment and LC-MS/MS approaches to quantifying the phosphoproteome. Given that most TAMs originate from the bone marrow, BMDMs have been established as a valid and robust model for evaluating cancer cell phagocytosis. CD47, a prominent “don’t eat me” signal expressed on various cancer cells [12,13], interacts with its receptor Sirpα on macrophages to inhibit phagocytosis [63]. We utilized a CD47-blocking antibody that disrupts the CD47-Sirpα interaction to induce cancer cell phagocytosis by the macrophages. Compared to several other commonly used strains, the binding affinity between BALB/c Sirpα and human CD47 is comparable to that between the human Sirpα and CD47 pair [64]. Therefore, we generated BMDMs using bone marrow cells from BALB/c mice. A human non-Hodgkin’s lymphoma cell line, Raji cells, was used as the target cancer cells. The BMDMs and Raji cells were co-cultured for 0 min, 1 min, 15 min, 60 min, and 120 min in the presence of CD47-blocking antibodies, with biological replicates. After the designated incubation periods, the Raji cells were washed off, and the macrophage proteins were digested into peptides, labeled with different TMT tags, pooled, and subjected to SMOAC for phospho-enrichment using subsequent TiO_2_ and IMAC columns (Figure 1A).

The Principal Component Analysis (PCA) of the phosphoproteome datasets demonstrated strong reproducibility across the experimental replicates, confirming the robustness of our data (Appendix A). To effectively differentiate between experimental variability and biological changes, we analyzed replicate samples collected at 0 and 120 min. In the null comparisons (replicate samples under the same condition), the distribution of the biological replicates appeared to be random (Figure 1B, left). However, comparisons across these time points revealed clear differences between early- and late-stage macrophage phagocytosis, with high consistency maintained among replicates (Figure 1B, right and Appendix A). Additional analyses, such as partial least squares–discriminant analysis (PLS-DA) of the differentially abundant (DA) phosphosites (Figure 1C) and clustering of these phosphosites (Figure 1D), further validated the reproducibility of our results. In the phosphoproteome profiling, we quantified a total of 11,658 phosphosites from the macrophage cells. Using a time series analysis with the limma package in R, we identified 3060 significant DA phosphosites across time (q-value < 0.05) (Figure 1D). The PLS-DA and heatmap analyses of the differentially expressed phosphosites demonstrated that the phosphoproteome started to change within minutes after the initiation of cancer cell phagocytosis. These results revealed distinct signaling events at the initial stages of phagocytosis (0 and 1 min time points) compared to those during the sustentation stages (Figure 1C,D).

### 3.2. Phosphoproteome Profiling Identifies Activation of the mTOR Pathway During Macrophage Phagocytosis

To gain more insights into the regulation of macrophage phagocytosis by biological pathways, we applied KEGG pathway enrichment to identify the top 20 pathways based on the significant DA phosphosites at different time points. This approach allowed us to understand the functional implications of these alterations in the cellular response to phagocytosis. Our analysis revealed significant enrichment in pathways including MAPK signaling, regulation of the actin cytoskeleton, autophagy and mTOR signaling, and FcγR-mediated phagocytosis (Figure 2A).

Subsequently, we employed the machine learning algorithm IKAP to infer the kinase activity based on the substrate phosphorylation levels derived from the phosphoproteome [54]. This analysis delineated the activities of 96 kinases, 22 of which were found to be differentially regulated and hierarchically clustered into two principal groups (Appendix A). The majority of the upregulated kinases have been reported to play critical roles in the regulation of cell cycle progression, cellular stress responses, and survival signaling, particularly within the MAPK, CDK, AMPK, and mTOR pathways. Notably, RPS6KA1 and RPS6KB2, which are downstream components of the mTOR pathway, exhibited sustained upregulation from 15 min to 120 min. Additionally, other kinases involved in growth factor signaling, such as MAP3K8, MARK2, and MAPK3 within the ERK/MAPK signaling cascade, were also upregulated following macrophage phagocytosis. CDK2 and CDK7, essential regulators of the cell cycle, were among those highlighted in this study.

We then performed a DEGPattern analysis for the DA phosphosites to identify clusters exhibiting highly correlated abundance patterns over time. This analysis revealed seven distinct clusters, with unique temporal abundance patterns of the DA phosphosites. Clusters 1, 4, and 5 displayed downregulated patterns, while clusters 2, 3, 6, and 7 displayed upregulated patterns (Figure 2B). Nucleocytoplasmic transport was significantly enriched in all three clusters, with increasing trends after 1 min. Cluster 2 was predominantly associated with proteins involved in mRNA processing, encompassing pathways such as the spliceosome, ATP-dependent chromatin remodeling, and mRNA surveillance. Clusters with an increasing phosphosite abundance during phagocytosis showed enrichment in MAPK signaling, autophagy, and cell cycle-related activities. Cluster 7 did not exhibit any significantly enriched pathways. FcγR-mediated phagocytosis was highly enriched in both cluster 1 and cluster 6, indicating that the components in this pathway were involved in both the early and late stages of phagocytosis (Figure 2C).

Cluster 3 comprised phosphosites that were relatively scarce in the early stages following the initiation of phagocytosis but exhibited a steady increase from 15 min onward, reflecting a pattern of temporal abundance of candidate signaling pathways potentially involved in the sustentation of phagocytosis (Figure 2B). Notably, the mammalian target of rapamycin (mTOR) signaling pathway is significantly enriched in cluster 3 (Figure 2C). mTOR signaling is frequently activated in cancer and plays critical roles in regulating the proliferation and metabolism of cancer cells [65]. Emerging data suggest that mTOR signaling is altered in approximately 30% of cancers [66]. The depletion of TSC1, a negative regulator of mTORC1, has been shown to lead to TAM reprogramming and the suppression of tumor growth [67]. We investigated the expression of the mTOR pathway components in the TME by analyzing single-cell RNA sequencing (scRNA-seq) data from MC38 colorectal tumors and HER2+ breast tumors sourced from the GEO database (GSE180296 and GSE139492). Distinct gene expression patterns defined clusters corresponding to various tumor-infiltrating immune cell types (Figure 2D,E and Appendix A). Notably, we identified a relatively higher enrichment of the mTOR pathway in tumor-infiltrating myeloid cells (monocytes/macrophage) compared to the other immune cell clusters, supporting the notion that mTOR signaling may play a role in modulating the activity of these cells within the TME (Figure 2F,G).

To functionally characterize the role of the mTOR pathway in macrophage phagocytosis, we employed rapamycin, a specific mTOR inhibitor, to treat the macrophages and assess its effect. The primary BMDMs were pretreated with various doses of rapamycin for 24 h and then co-cultured with Raji cells transduced with luciferase–eGFP for an additional 24 h in the absence or presence of the CD47-blocking antibody. A luminescence-based phagocytosis assay was utilized to evaluate the phagocytic capability of the macrophages by quantifying the luminescence signals from the surviving cancer cells [49]. We showed that the rapamycin treatment elicited the inhibition of Raji cell phagocytosis in a dose-dependent manner (Figure 2H) and aligned with different E:T ratios (Figure 2I). Consistently, robust inhibition of phagocytosis by rapamycin was observed in the primary human peripheral blood monocyte-derived macrophages (Figure 2J). The inhibitory effect of rapamycin on phagocytosis was further validated using two additional human cell lines—SW620 (human colon cancer) and MDA-MB-231 (human breast cancer) (Appendix A)—suggesting that this mechanism may be broadly applicable across multiple human cancers. In line with these results, treatment with temsirolimus (CCI-779), an alternative mTOR inhibitor that binds to FKBP-12 [68], produced comparable effects in inhibiting the phagocytosis of the Raji cells by the macrophages (Appendix A). Lastly, we performed genetic suppression by knocking down the expression of MTOR in the macrophages. We showed that genetic suppression of MTOR significantly reduced tumor cell phagocytosis compared to that in the control knockdown group, consistent with the results of the pharmacological inhibition of the mTOR pathway (Appendix A).

### 3.3. Blockade of the mTOR Pathway Shows No Impact on the Cell Surface Receptor Expression or Polarization States of Macrophages

To determine whether the reduced phagocytic ability of the macrophages following rapamycin treatment was attributable to cytotoxic effects on the macrophages, flow cytometry was employed to assess the cell viability using Annexin V and nucleic acid staining. Our analysis indicated that rapamycin treatment did not compromise macrophage viability at the dose at which phagocytosis was inhibited (Figure 3A).

Fcγ receptors (FcγRs) are the principal receptors on macrophages responsible for binding the Fc region of antibodies and mediating antibody-dependent cellular phagocytosis (ADCP) [69]. Downregulation of cell surface Fc receptors during hypophagia has been identified as a major mechanism contributing to diminished phagocytic activity [70]. Therefore, we investigated whether mTOR pathway inhibition affected the expression of FcγRs on the macrophages. Our results demonstrated that the expression levels of all of the cell surface FcγRs, including activating FcγRs (FcgRI, FcgRIII, and FcgRIV) and inhibitory FcγRs (FcgRIIB) [71], remained unchanged following 24 h of rapamycin pretreatment (Figure 3B), suggesting that rapamycin does not impair macrophages’ ability to recognize and bind to antibody-coated cancer cells. In addition, the expression of Sirpα on the macrophages’ cell surfaces was unaffected by the rapamycin treatment (Figure 3C). Collectively, these findings indicate that mTOR signaling blockade does not directly impair target cell recognition by macrophages. This is consistent with our phosphoproteomics data, which showed that mTOR signaling was not immediately activated during the initial stage but was progressively amplified during the sustained phagocytic response.

We next evaluated the potential impact of rapamycin on the phenotypic polarization of the macrophages by analyzing the cell surface expression of markers associated with M1- and M2-like polarization states. Our results indicated that rapamycin, at the dose at which it significantly inhibited phagocytosis, had minimal effects on the expression of either M1 (CD80, CD86, iNOS, and MHC II) or M2 polarization markers (CD206) (Figure 3D).

### 3.4. Blockade of mTOR Signaling Impairs the Capacity of Macrophages for Phagocytosis but Not Their Affinity

We further investigated the underlying mechanisms of mTOR signaling in macrophage phagocytosis. The current routes of antibody administration frequently do not achieve saturation of antigen binding at local tumor sites [72]. To recapitulate the conditions of the TME best with antibody therapy, we utilized varying doses of CD47-blocking antibodies, with concentrations of up to 0.2 µg/mL representing partial CD47 binding, while concentrations above 1 µg/mL represented saturated CD47 binding (Appendix A).

Three types of phagocytosis assays were employed to evaluate the effects and mechanisms of mTOR blockade on tumor cell phagocytosis: (1) flow-cytometry- and (2) microscopy-based phagocytosis assays, to assess the phagocytic affinity by quantifying the macrophages capable of engulfing tumor cells, and (3) luminescence-based phagocytosis assays, to assess the phagocytic capacity by quantifying the tumor cells that survived phagocytosis.

In the first experiment, the BMDMs were treated with control vehicle or rapamycin and then co-cultured with the Raji cells for 15 min, 30 min, and 90 min or 2 h in the presence of the CD47-blocking antibody before being subjected to a flow cytometry analysis or microscopy visualization. In both assays, in line with our observations that mTOR blockade had a minimal impact on the cell surface machinery mediating the macrophage–cancer cell interaction, there were no significant differences in phagocytic affinity (the percentage of macrophages phagocytosing the Raji cells) between the control- and rapamycin-pretreated macrophages at any of the time points assessed (Appendix A). We then used flow cytometry to examine the macrophages’ phagocytic ability by extending the co-culture period with the Raji cells to 2 h, 6 h, 12 h, or 24 h in the presence of low (0.02 µg/mL), medium (0.2 µg/mL), or high (2 µg/mL) concentrations of the CD47-blocking antibody. No significant differences in the phagocytosis of the Raji cells between the control- and rapamycin-treated macrophages were observed under any of these conditions (Figure 4A–C).

Lastly, we evaluated the effects of mTOR blockade on the macrophages’ phagocytic capacity by quantifying the number of surviving Raji cells after co-culturing them with the control- or rapamycin-treated BMDMs for 2 h, 6 h, 12 h, or 24 h. Reduced phagocytosis by the rapamycin-treated macrophages was detected starting at 12 h of co-culture in the presence of both 0.02 and 0.2 µg/mL of the CD47-blocking antibody (Figure 4D).

Taken together, these data suggested that mTOR blockade using rapamycin impaired the capacity, but not the affinity, of macrophages in phagocytosis of target cancer cells.

### 3.5. Blockade of mTOR Signaling Delays Phagosome Maturation and Compromises Non-Phagocytic Macrophages’ Transition into a Phagocytic State with Cancer Cell Rechallenge

During phagocytosis, the maturation of phagolysosomes requires the fusion of phagosomes with lysosomes, which results in phagolysosomes and the degradation of the digested particles [73]. To determine whether the impaired capacity of the macrophages for phagocytosis when using rapamycin was due to dysfunction of the phagolysosomes via defective lysosome acidification, we utilized LysoTracker staining to monitor the acidity of the phagosomes in the macrophages [74]. The fluorescence intensity of LysoTracker correlates with the acidity of lysosomes and phagosomes. Following co-culture with the Raji cells for various time points, ranging from 2 h to 24 h, the control- or rapamycin-treated macrophages were stained with LysoTracker and subjected to flow cytometry to assess phagosome maturation. We observed a significant increase in the acidity of the phagosomes in the macrophages starting at 6 h post-co-culture with the Raji cells (Figure 5A,B). Phagocytic macrophages that had engulfed the Raji cells exhibited a stronger intensity of LysoTracker staining compared to the bystander non-phagocytic macrophages (Figure 5A,B and Appendix A). Notably, the enhanced acidity of the phagosomes observed during phagocytosis was largely inhibited in the macrophages by mTOR blockade (Figure 5A,B and Appendix A), corresponding with their reduced phagocytosis capacity (Figure 4D).

Sustained phagocytic signaling elicited continuous phagocytosis of multiple target cancer cells (Appendix A). To investigate how the initial engulfment influenced subsequent phagocytic events, we conducted a rechallenge experiment using Raji cells labeled with CFSE (green, termed G-Raji) or CellTrace Far Red dye (Far Red, termed FR-Raji) as target cells. The macrophages were co-cultured with the G-Raji cells in the presence of the CD47-blocking antibody for 6 h or 12 h. Subsequently, the G-Raji cells were removed, and the macrophages were rechallenged with the FR-Raji cells for an additional 18 h or 12 h. The proportion of double-positive cells, the macrophages that had engulfed both the G-Raji cells and the rechallenged FR-Raji cells, was found to be larger compared to that of the single-positive cells, the macrophages that had only phagocytosed either the initial G-Raji cells or the rechallenged FR-Raji cells (Figure 5C,D). This suggested that initial engulfment of a target cell did not trigger rapid attenuation of the macrophages’ phagocytic affinity. Importantly, while the rapamycin treatment showed no impact on the phagocytosis of the G-Raji cells, it resulted in a moderate reduction in the percentage of macrophages that engulfed both the G-Raji cells and FR-Raji cells. A marked reduction in the percentage of macrophages that solely phagocytosed the FR-Raji cells was observed following rapamycin treatment (Figure 5C,D). Together, these findings indicated that mTOR blockade impaired the sustentation and amplification of phagocytosis, leading to a reduction in the continuous phagocytosis of the rechallenged target cells and compromised transition of the macrophages from a non-phagocytic to a phagocytic state.

## 4. Discussion

Exploiting the phagocytic activity of tumor-associated macrophages (TAMs) represents an innovative strategy for cancer therapy. However, the precise signaling networks that drive macrophages during the phagocytosis of cancer cells remain inadequately characterized [75]. In this study, we have shown that the dynamic phagocytosis process consists of distinct early and late phases that involve extensive phosphoproteomic changes. Phosphoproteomic profiling affords more immediate and intricate insights into the activation/deactivation states of the signaling pathways in macrophages during the phagocytosis of cancer cells [76]. This methodology allows for a direct analysis of the post-translational modifications in functional proteins, thereby offering a more exhaustive understanding of cellular function and regulation beyond the transcriptional level [77,78]. Advancements in mass-spectrometry-based techniques have significantly enhanced the scope and precision of phosphoproteomic profiling [78,79]. Using phosphoproteomic data, this study seeks to uncover novel insights into the basic mechanisms of macrophage-mediated phagocytosis of cancer cells [80,81,82]. We characterized the temporal phosphorylation dynamics of the signaling pathways within macrophages that are critical to sustaining tumor cell phagocytosis. Our work provides a valuable resource on the phosphorylation events associated with phagocytosis by macrophages.

Phagocytosis is accompanied by rapid and transient acidification of the phagosomes [73]. mTORC1 has been reported to regulate the expression of V-ATPases, which are responsible for transporting protons into the lysosome lumen or out of cells into the extracellular medium, in a TFEB-dependent manner [83]. The increased assembly of V-ATPases during the maturation of bone-marrow-derived dendritic cells was inhibited by rapamycin [84]. Furthermore, in macrophages and dendritic cells, mTOR is required for lysosome tabulation and antigen presentation induced by LPS [85]. Interestingly, while our findings suggest that mTOR blockade using rapamycin has minimal effects on steering BMDMs toward M1- or M2-like phenotypes, the induction of M1 or M2 polarization by LPS or IL4 may still involve mTOR activities, exemplified by the findings that constitutive mTORC1 activity impairs M2 polarization [42] and rapamycin shifts the polarization of human macrophages toward the M1 phenotype [86]. M1-like macrophages were reported to delay the acidification of phagosomes, while M2-like macrophages may facilitate immediate phagosome acidification [87,88]. In our study, we demonstrated that mTOR blockade using rapamycin displayed minor effects on the acidity of the lysosomes in resting macrophages, as indicated by LysoTracker staining. However, once they were incubated with target cancer cells, a significant enhancement in lysosome and phagosome acidification was observed in the macrophages, which was effectively blocked by rapamycin. These findings highlight the multifaceted roles of mTOR pathways in regulating lysosome activities critical to various cellular functions. The molecular mechanisms underlying phagosome maturation during cancer cell phagocytosis, as well as its regulation by mTOR signaling to sustain phagocytosis, warrant further exploration. Notably, lysosomal degradation is essential for sustained phagocytosis of bacteria by macrophages, mediated by the endolysosomal Cl- transporter ClC-b/CLCN7, with NF-κB signaling coupling phagolysosomal degradation to sustained clearance of bacteria [89]. Future research should investigate whether and how mTOR signaling impacts the ClC-b/CLCN7 and NF-κB pathways in macrophage-mediated phagocytosis of cancer cells.

Recent studies have shown that enhancing mTORC1 signaling drives a pro-resolving macrophage state that inhibits tumor growth [67], underscoring an innate-immunity-mediated tumor suppression mechanism with potential applications in cancer immunotherapy. Distinct from therapies that primarily target adaptive immunity or the tumor cells directly, our study advances our understanding of the fundamental principles of mTOR signaling that govern macrophage-mediated tumor cell phagocytosis. This knowledge could be harnessed to develop novel immunotherapeutic strategies targeting macrophages, which are often predominantly immune components in tumors, offering promising treatment options for a wide range of cancers.

## 5. Conclusions

In summary, we have delineated the phosphoproteomic landscape of macrophage phagocytosis and identified the activation of the mTOR pathway during this process. Our findings indicate that mTOR inhibition impairs the capacity of macrophages for phagocytosis but not their affinity with it and delays phagosome acidification during phagocytosis, impeding the sustentation of phagocytosis and hindering the transition of non-phagocytic macrophages to phagocytic states. Mechanistic insights into macrophage-mediated cancer cell phagocytosis may inspire novel therapeutic strategies for cancer treatment.

## Figures and Tables

**Figure 1 cancers-16-04238-f001:**
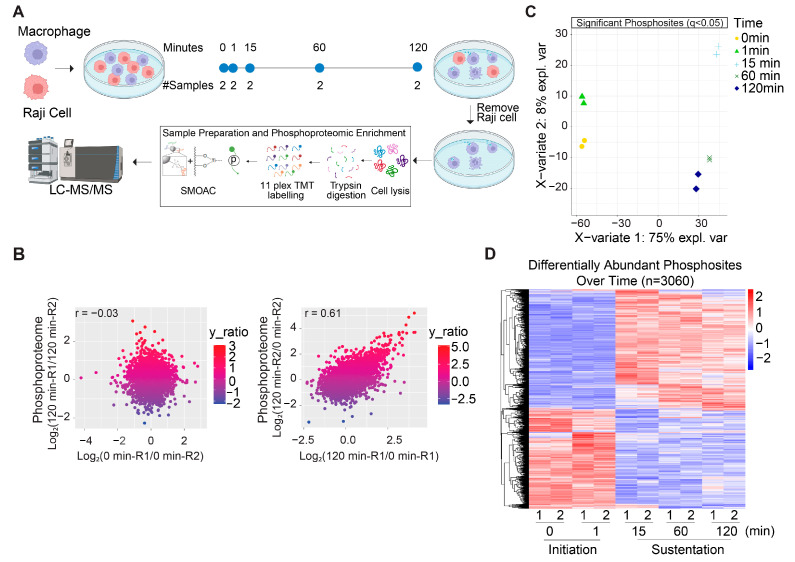
A multiplexed quantitative analysis of the phosphoproteome was applied to analyzing phagocytosis of cancer cells by macrophages. (**A**) A schematic showing the design of the phosphoproteomic profiling. BMDMs were co-cultured with Raji cells, and the Raji cells were removed 0, 1, 15, 60, or 120 min after the co-culture. The macrophages were then lysed, and total proteins were digested into peptides and labeled with tandem mass tags (TMTs) before being pooled. The samples were then subjected to SMOAC phospho-enrichment, and phosphopeptides were detected using an Orbitrap Fusion Lumos Tribrid MS. (**B**) Null comparisons of the phosphoproteome datasets (0 min-R1 vs. 0 min-R2 and 120 min-R1 vs. 120 min-R2) (**left**) show significant differences in the patterns when contrasted with the true comparisons (120 min-R1 vs. 0 min-R1 and 120 min-R2 vs. 0 min-R2) (**right**). Each dot represents a phosphosite, with the colors indicating the log2 fold change ratio (y_ratio). The correlation coefficient (r) for each comparison is provided, with R signifying a replicate. (**C**) Partial least squares–discriminant analysis (PLS-DA) showing clustering of the time point replicates from the macrophage co-culture from the 3060 significant DA phosphosites. (**D**) Heatmap showing unsupervised clustering of the DA phosphosites, detected via a time series analysis using a linear model through limma. Clustering was calculated using Pearson’s correlation coefficient. The *x*-axis indicates the time points (0, 1, 15, 60, and 120 min), with the replicates for each condition. The *y*-axis represents individual phosphosites, grouped by the similarity in their abundance patterns throughout the time series, while the color gradient depicts the z-scores (scaled phosphosite abundance).

**Figure 2 cancers-16-04238-f002:**
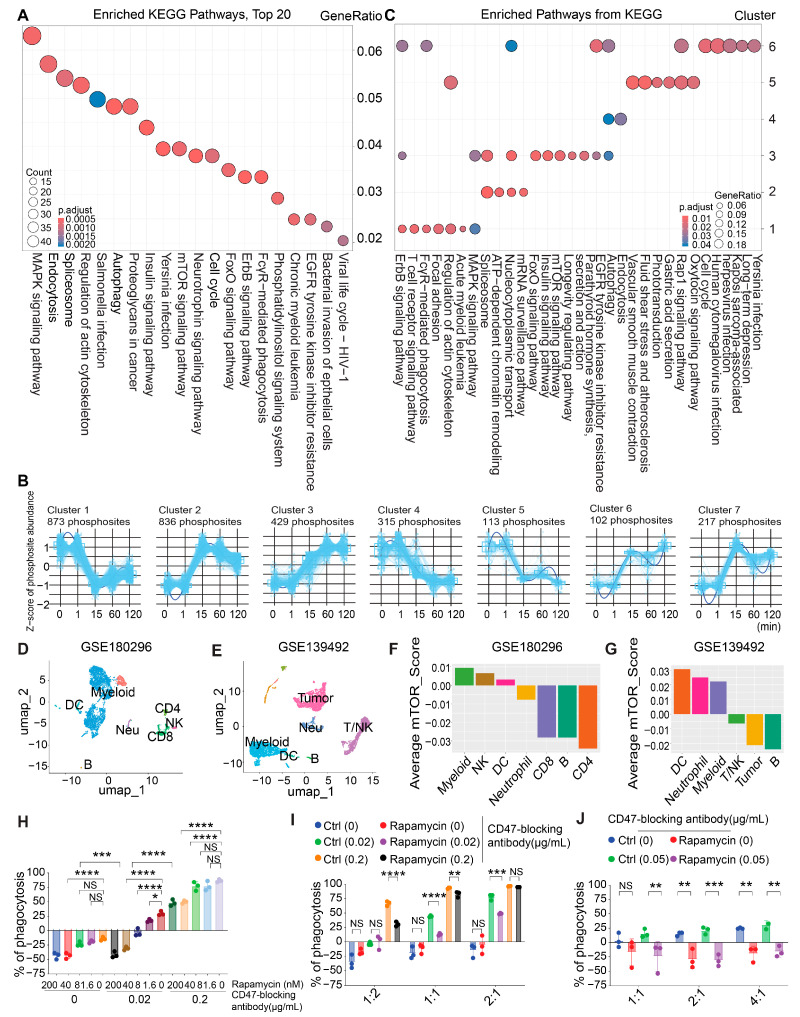
Phosphoproteome profiling identified the activation of the mTOR pathway during macrophage phagocytosis. (**A**) Top 20 pathways with differentially abundant phosphosites according to KEGG pathway analysis. The circle size represents the count of differentially abundant phosphosites, and the color indicates the adjusted *p*-value (p.adjust). The *x*-axis displays the pathway names, while the *y*-axis represents the GeneRatio (the proportion of genes in each pathway relative to the total number of genes analyzed). (**B**) Clusters of phosphosites with similar trends were identified using a DEGPattern analysis. Only patterns with at least 50 phosphosites were retained, resulting in 7 final clusters. (**C**) Dot plots of significantly enriched pathways for each cluster (except cluster 7, which reported no significant pathways). The dot color indicates the *p*-value of enrichment, while the circle size represents the ratio of genes from the pathway present in each cluster. The *x*-axis displays the pathway names, while the *y*-axis represents the cluster numbers. (**D**,**E**) UMAP plots showing immune cell clusters of colorectal and breast tumors (GSE180296 and GSE139492). (**D**) The cluster includes myeloid cells, NK cells, DCs, neutrophils, and CD8, B, and CD4 cells. (**E**) The cluster includes DCs, neutrophils, and myeloid, T/NK, tumor, and B cells. (**F**,**G**) The bar plot illustrates the average mTOR pathway activity (mTOR_Score) across different cell types. (**H**) Rapamycin treatment significantly inhibited CD47-blockade-induced phagocytosis of the Raji cells (200 nM, 40 nM, 8 nM, and 1.6 nM rapamycin) by the BMDMs. N = 3. (**I**) Rapamycin pretreatment (100 nM) significantly inhibited phagocytosis of the Raji cells by the BMDMs at different E:T ratios. N = 3. (**J**) Rapamycin treatment (100 nM) significantly inhibited CD47-blockade-induced phagocytosis of the Raji cells at different E:T ratios by human peripheral blood monocyte-derived macrophages. N = 3. Data are represented as means ± SDs. NS indicates not statistically significant; * *p* < 0.05, ** *p* < 0.01, *** *p* < 0.001, and **** *p* < 0.0001, as determined by one-way ANOVA (H) or two-way ANOVA (I, J).

**Figure 3 cancers-16-04238-f003:**
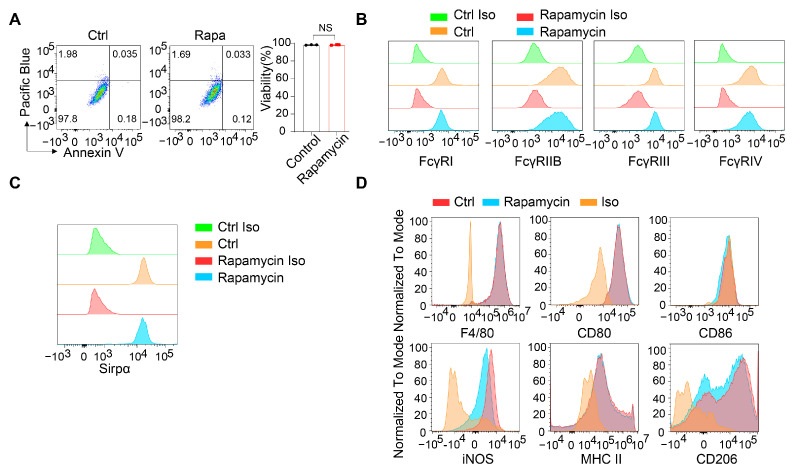
Blockade of the mTOR pathway shows no impact on the cell surface receptor expression or polarization states of the macrophages. (**A**) Cell viability of the macrophages was analyzed at 24 h with or without 100 nM rapamycin treatment. N = 3. (**B**) Representative FACS plots showing the expression of FcүRI, FcүRIIB, FcүRIII, and FcүRIV on the BMDMs with or without 100 nM rapamycin treatment for 24 h. N = 3. (**C**) Representative FACS plots showing the expression of SIRPα on the BMDMs with or without 100 nM rapamycin treatment for 24 h. N = 3. (**D**) Representative FACS plots showing the expression of F4/80, CD80, CD86, iNOS, MHC II, and CD206 on the BMDMs with or without 100 nM rapamycin pretreatment for 24 h. N = 3. Data are represented as means ± SD. NS indicates not statistically significant as determined using an unpaired t-test (**A**).

**Figure 4 cancers-16-04238-f004:**
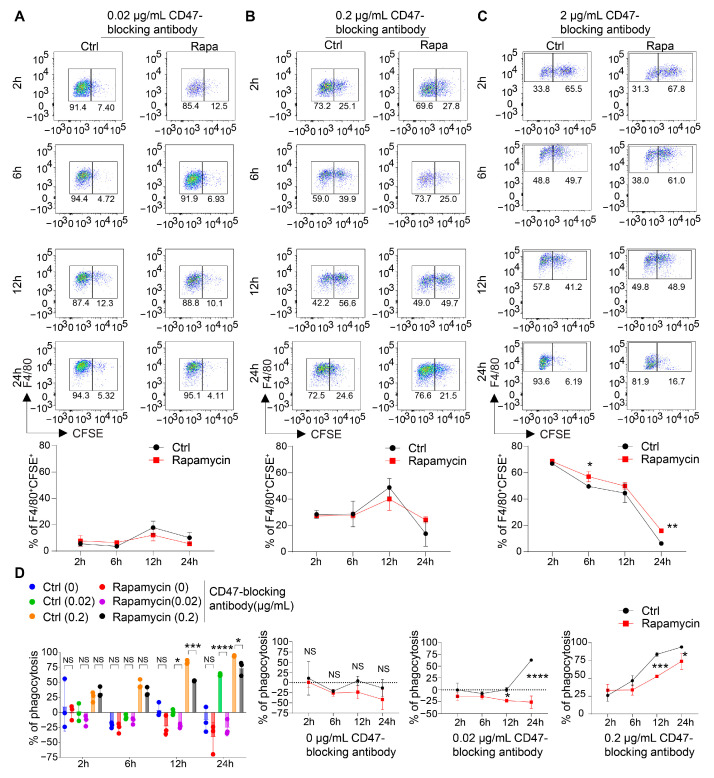
Blockade of mTOR signaling impairs the capacity of macrophages for phagocytosis but not their affinity with it. (**A**–**C**) A flow-cytometry-based phagocytosis assay. CFSE-labeled Raji cells were incubated with BMDMs treated with control vehicle or rapamycin in the presence of 0.02, 0.2, or 2 µg/mL of CD47-blocking antibody. Representative FACS plots showing the expression of F4/80^+^CFSE^+^ cells at 2 h, 6 h, 12 h, and 24 h in the control group and the rapamycin group. The percentages of F4/80^+^CFSE^+^ cells at different time points were summarized for all three groups. N = 3. (**D**) A luminescence-based phagocytosis assay. Raji cells were incubated with BMDMs treated with control vehicle or rapamycin in the presence of 0, 0.02, or 0.2 µg/mL of CD47-blocking antibody. Surviving Raji cells were quantified using luminescence. N = 3. Data are represented as means ± SDs. NS indicates not statistically significant; * *p* < 0.05, ** *p* < 0.01, *** *p* < 0.001, and **** *p* < 0.0001, as determined using two-way ANOVA (**A**–**D**).

**Figure 5 cancers-16-04238-f005:**
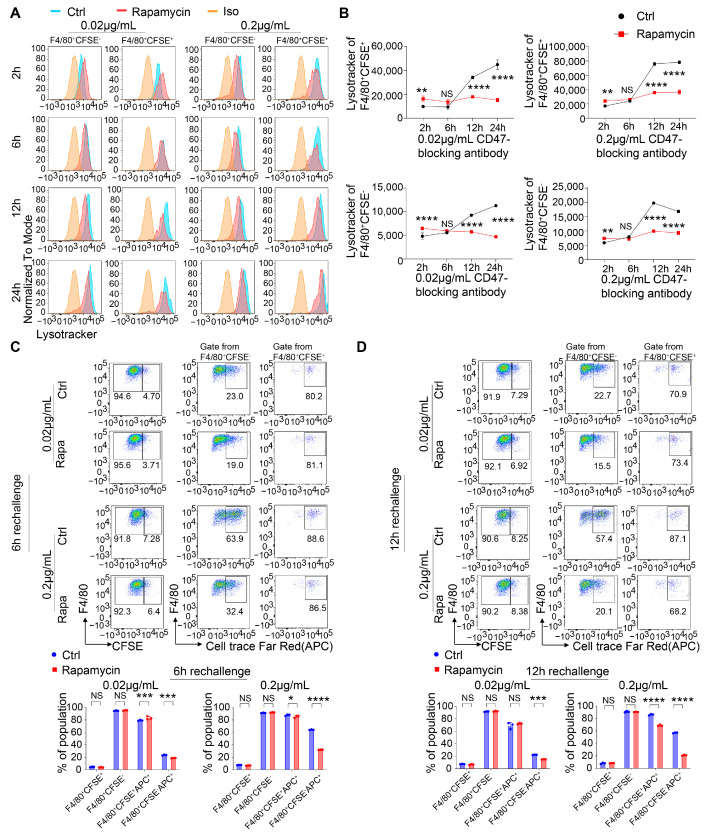
Blockade of mTOR signaling delays phagosome maturation and compromises non-phagocytic macrophages’ transition into a phagocytic state with cancer cell rechallenge. (**A**,**B**) LysoTracker staining of the BMDMs after co-culture with the Raji cells in the presence of 0.02 or 0.2 µg/mL of CD47-blocking antibody. BMDMs were treated with control vehicle or rapamycin. (**A**) Representative FACS plots. The cells were assessed using FACS to quantify the intensity of the lysosome acidity of the F4/80^+^CSFE^+^ and F4/80^+^CSFE^−^ populations. (**B**) Summary of the Mean Fluorescence Intensity (MFI) of LysoTracker gated from the F4/80^+^CSFE^+^ and F4/80^+^CSFE^−^ cells. N = 3. (**C**,**D**) BMDMs were co-cultured with CFSE-labeled Raji cells and rechallenged with Far Red-labeled Raji cells at 6 h or 12 h and then harvested at 24 h in the presence of 0.02 or 0.2 µg/mL of CD47 blocking antibody. (**C**) Representative FACS plots show the percentage of the F4/80^+^APC^+^ population gated from the F4/80^+^CSFE^+^ or F4/80^+^CSFE^−^ population and a summary of the percentages of the F4/80^+^CSFE^+^, F4/80^+^CSFE^−^, F4/80^+^CSFE^+^APC^+^, and F4/80^+^CSFE^−^APC^+^ populations in the 6 h (**C**) and 12 h rechallenge experiments (**D**). N = 3. Data represented as means ± SDs. NS indicates not statistically significant; * *p* < 0.05, ** *p* < 0.01, *** *p* < 0.001, and **** *p* < 0.0001, as determined using two-way ANOVA (**B**–**D**).

## Data Availability

The data will be made available upon reasonable request to the corresponding authors.

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
