# Peer review of "Phosphoproteomic Profiling Reveals mTOR Signaling in Sustaining Macrophage Phagocytosis of Cancer Cells"

_cancers, 2024, doi:10.3390/cancers16244238_

Round 1
Reviewer 1 Report
Comments and Suggestions for Authors
-
The abstract is comprehensive but could be more concise. Highlight the core findings and their implications briefly to enhance readability.
-
Add a more detailed context about why targeting mTOR signaling in macrophages is innovative compared to existing cancer therapies.
-
Figures such as heatmaps or pathway enrichments could include clearer legends and larger text for better understanding.
-
Provide more specifics about statistical tools (e.g., why certain thresholds like q-value < 0.05 were chosen) to ensure reproducibility.
-
Summarize key quantitative findings in tables alongside graphs for easy reference.
-
Consider breaking down sections on phosphoproteomic profiling further into initiation, sustenance, and resolution phases for better logical flow.
-
Elaborate on how the findings could translate to clinical settings, particularly in developing macrophage-based therapies.
-
Strengthen the introduction and discussion with more references to similar phosphoproteomic studies.
-
Simplify complex sentences, especially in technical descriptions, to ensure broader accessibility.
-
Check for uniformity in citations, terminologies (e.g., BMDMs vs. bone marrow-derived macrophages), and formatting styles throughout the manuscript.
Author Response
- The abstract is comprehensive but could be more concise. Highlight the core findings and their implications briefly to enhance readability.
Thank you for your valuable suggestion. We have revised the abstract accordingly.
- Add a more detailed context about why targeting mTOR signaling in macrophages is innovative compared to existing cancer therapies.
Thank you for this insightful suggestion. The role of mTOR signaling in macrophage phagocytosis as a component of innate immunity remains insufficiently understood. Given the significant therapeutic potential of macrophage-mediated cancer cell phagocytosis, a mechanistic investigation into the role of the mTOR pathway in macrophages could provide valuable insights for developing therapeutic interventions. We have incorporated the rationale highlighting the innovative role of mTOR signaling in macrophages into the introduction and discussion sections.
- Figures such as heatmaps or pathway enrichments could include clearer legends and larger text for better understanding.
The legends and text of figures have been revised accordingly (Figure 1D and 2A,2C.).
- Provide more specifics about statistical tools (e.g., why certain thresholds like q-value < 0.05 were chosen) to ensure reproducibility.
To identify differentially abundant phosphosites over time, we utilized the lmFit and eBayes functions from the limma package in R (v3.5.2). P-values were adjusted with a Benjamini-Hochberg correction for multiple testing (q-value), with a significance threshold set at q-value <0.05. This threshold was chosen to balance sensitivity (detecting relevant changes) and specificity (minimizing false positives), aligning with commonly accepted standards to ensure robust and reproducible findings across similar datasets. Additionally, we have also included details of the R version (v3.5.2) and parameter settings for the reproducibility of the analysis by other researchers.
- Summarize key quantitative findings in tables alongside graphs for easy reference.
Thank you for the suggestion. Tables summarizing key quantitative findings have been included in supplemental Table 1.
- Consider breaking down sections on phosphoproteomic profiling further into initiation, sustenance, and resolution phases for better logical flow.
- Elaborate on how the findings could translate to clinical settings, particularly in developing macrophage-based therapies.
Thank you for the advice. We have incorporated the discussions accordingly. Our study provides a significant advancement in understanding the fundamental principles of mTOR signaling that regulate macrophage-mediated tumor cell phagocytosis. This knowledge could pave the way for leveraging novel innate immune anti-tumor mechanisms to develop highly effective immunotherapeutic strategies applicable across multiple cancer types.
- Strengthen the introduction and discussion with more references to similar phosphoproteomic studies.
Thank you for the suggestion. Additional references related to phosphoproteomic studies have been incorporated into the introduction section. Furthermore, we have emphasized the significance of our phosphoproteomic data in the discussion section to provide greater context and relevance to our findings.
- Simplify complex sentences, especially in technical descriptions, to ensure broader accessibility.
Thank you for the suggestion. We have thoroughly reviewed the text and simplified complex sentences, particularly in technical sections, to ensure the content is more accessible to a broader audience.
- Check for uniformity in citations, terminologies (e.g., BMDMs vs. bone marrow-derived macrophages), and formatting styles throughout the manuscript.
The uniformity of citations, terminologies, and formatting styles has been examined and confirmed.
Reviewer 2 Report
Comments and Suggestions for Authors
1. Title: The title should be capitalized appropriately, concise, descriptive, and accurate to ensure readers can easily discover and cite the work by searching relevant keywords.
2. Abstract: The abstract needs to be rewritten to clearly emphasize the main content and novel aspects of the study, which are currently not evident.
3. Introduction: Expand the introduction to better highlight the study's novelty and significance, particularly in comparison to previous research.
4. Provide some digital captures for clarity and verification.
5. Figure Resolution: Improve the resolution of Figures 1, 2, 3a, 4, and 5 to ensure enhanced clarity.
6. Supplementary File: Increase the resolution of Figures 3 and 4 included in the supplementary file.
7. Comparison and Outlook: Compare the study's findings with those of existing research to highlight similarities and differences. Expand on the implications for future research, and consider relocating this discussion to precede or be incorporated into the conclusion section.
8. Manuscript Formatting: The entire manuscript's format needs a thorough review. This includes renaming headings and subheadings to improve clarity and consistency.
9. References: The reference list should be revised to comply with the journal's formatting requirements. Furthermore, additional references related to the Cancers should be incorporated.
10. Language and Grammar: The manuscript's English grammar should be thoroughly reviewed and corrected to enhance readability and coherence.
Comments on the Quality of English LanguageLanguage and Grammar: The manuscript's English grammar should be thoroughly reviewed and corrected to enhance readability and coherence.
Author Response
- Title: The title should be capitalized appropriately, concise, descriptive, and accurate to ensure readers can easily discover and cite the work by searching relevant keywords.
Thank you for the suggestion. We have capitalized the title as recommended. The current title succinctly captures the innovative findings of our study and is designed to facilitate literature searches by readers.
- Abstract: The abstract needs to be rewritten to clearly emphasize the main content and novel aspects of the study, which are currently not evident.
Thank you for the suggestion. We have revised the abstract accordingly.
- Introduction: Expand the introduction to better highlight the study's novelty and significance, particularly in comparison to previous research.
Thank you for the thoughtful feedback. We have expanded the introduction to highlight the novelty of our approach in using phosphoproteomics to study macrophage phagocytosis, as well as the originality of our findings.
- Provide some digital captures for clarity and verification.
Thank you for the suggestion. Representative figures for single-cell RNA sequencing analysis and flow cytometry have been provided in Fig2-5.
- Figure Resolution: Improve the resolution of Figures 1, 2, 3a, 4, and 5 to ensure enhanced clarity.
Thank you for the suggestion. The resolution of figures has been improved accordingly in the revised manuscript.
- Supplementary File: Increase the resolution of Figures 3 and 4 included in the supplementary file.
Thank you for the suggestion. The resolution of figures has been improved accordingly in the revised manuscript.
- Comparison and Outlook: Compare the study's findings with those of existing research to highlight similarities and differences. Expand on the implications for future research, and consider relocating this discussion to precede or be incorporated into the conclusion section.
Thank you for the constructive feedback. We have incorporated the comparison and revised the outlook as suggested, and re-organized the introduction and discussion sections accordingly.
8.Manuscript Formatting: The entire manuscript's format needs a thorough review. This includes renaming headings and subheadings to improve clarity and consistency.
Thank you for the suggestion. We have re-formatted the manuscript as suggested to enhance clarity and consistency. The subheadings have been revised.
- References: The reference list should be revised to comply with the journal's formatting requirements. Furthermore, additional references related to the Cancersshould be incorporated.
We have revised the reference list to comply with the journal's formatting requirements and incorporated additional references.
- Language and Grammar: The manuscript's English grammar should be thoroughly reviewed and corrected to enhance readability and coherence.
Thank you for the suggestions. Professional language editing has been performed to ensure the readability of coherence of the manuscript.
Round 2
Reviewer 1 Report
Comments and Suggestions for Authors
The authors responded to comments, and the manuscript can be accepted now after an editorial check.
Comments on the Quality of English LanguageNone